# CliqueCNN: Deep Unsupervised Exemplar Learning

**Miguel A. Bautista**[*]**, Artsiom Sanakoyeu**[*]**, Ekaterina Sutter, Björn Ommer**
Heidelberg Collaboratory for Image Processing
IWR, Heidelberg University, Germany
`firstname.lastname@iwr.uni-heidelberg.de`

## Abstract

Exemplar learning is a powerful paradigm for discovering visual similarities in an unsupervised manner. In this context, however, the recent breakthrough in deep learning could not yet unfold its full potential. With only a single positive sample, a great imbalance between one positive and many negatives, and unreliable relationships between most samples, training of Convolutional Neural networks is impaired. Given weak estimates of local distance we propose a single optimization problem to extract batches of samples with mutually consistent relations. Conflicting relations are distributed over different batches and similar samples are grouped into compact cliques. Learning exemplar similarities is framed as a sequence of clique categorization tasks. The CNN then consolidates transitivity relations within and between cliques and learns a single representation for all samples without the need for labels. The proposed unsupervised approach has shown competitive performance on detailed posture analysis and object classification.

## 1 Introduction

Visual similarity learning is the foundation for numerous computer vision subtasks ranging from low-level image processing to high-level object recognition or posture analysis. A common paradigm has been category-level recognition, where categories and the similarities of all their instances to other classes are jointly modeled. However, large intra-class variability has recently spurred exemplar methods [15, 11], which split this problem into simpler sub-tasks. Therefore, separate exemplar classifiers are trained by learning the similarities of individual exemplars against a large set of negatives. The exemplar paradigm has been successfully employed in diverse areas such as segmentation [11], grouping [10], instance retrieval [2, 19], and object recognition [15, 5]. Learning similarities is also of particular importance for posture analysis [8] and video parsing [17].

Among the many approaches for similarity learning, supervised techniques have been particularly popular in the vision community, leading to the formulation as a ranking [23], regression [6], and classification [17] task. With the recent advances of convolutional neural networks (CNN), two-stream architectures [25] and ranking losses [21] have shown great improvements. However, to achieve their performance gain, CNN architectures require millions of samples of supervised training data or at least the fine-tuning [3] on large datasets such as PASCAL VOC. Although the amount of accessible image data is increasing at an enormous rate, supervised labeling of similarities is very costly. In addition, not only similarities between images are important, but especially between objects and their parts. Annotating the fine-grained similarities between all these entities is hopelessly complex, in particular for the large datasets typically used for training CNNs.

Unsupervised deep learning of similarities that does not requiring any labels for pre-training or fine-tuning is, therefore, of great interest to the vision community. This way we can utilize large

---

[*]Both authors contributed equally to this work.
Project on GitHub: `https://github.com/asanakoy/cliquecnn`

image datasets without being limited by the need for costly manual annotations. However, CNNs for exemplar-based learning have been rare [4] due to limitations resulting from the widely used softmax loss. The learning task suffers from only a single positive instance, it is highly unbalanced with many more negatives, and the relationships between samples are unknown, cf. Sec. 2. Consequentially, stochastic gradient descend (SGD) gets corrupted and has a bias towards negatives, thus forfeiting the benefits of deep learning.

**Outline of the proposed approach:** We overcome these limitations by updating similarities and CNNs. Typically at the beginning only a few, local estimates of (dis-)similarity are easily available, i.e., pairs of samples that are highly similar (near duplicates) or that are very distant. Most of the similarities are, however, unknown or mutually contradicting, so that transitivity does not hold. Therefore, we initially can only gather small, compact cliques of mutually similar samples around an exemplar, but for most exemplars we know neither if they are similar nor dissimilar. To nevertheless define balanced classification tasks suited for CNN training, we formulate an optimization problem that builds training batches for the CNN by selecting groups of compact cliques, so that all cliques in a batch are mutually distant. Thus for all samples of a batch (dis-)similarity is defined—they either belong to the same compact clique or are far away and belong to different cliques. However, pairs of samples with no reliable similarities end up in different batches so they do not yield false training signal for SGD. Classifying if a sample belongs to a clique serves as a pretext task for learning exemplar similarity. Training the network then implicitly reconciles the transitivity relations between samples in different batches. Thus, the learned CNN representations impute similarities that were initially unavailable and generalize them to unseen data.

In the experimental evaluation the proposed approach significantly improves over state-of-the-art approaches for posture analysis and retrieval by learning a general feature representation for human pose that can be transferred across datasets.

## 1.1 Exemplar Based Methods for Similarity Learning

The Exemplar Support Vector Machine (Exemplar-SVM) has been one of the driving methods for exemplar based learning [15]. Each Exemplar-SVM classifier is defined by a single positive instance and a large set of negatives. To improve performance, Exemplar-SVMs require several round of hard negative mining, increasing greatly the computational cost of this approach. To circumvent this high computational cost [10] proposes to train Linear Discriminant Analysis (LDA) over Histogram of Gradient (HOG) features [10]. LDA whitened HOG features with the common covariance matrix estimated for all the exemplars removes correlations between the HOG features, which tend to amplify the background of the image.

Recently, several CNN approaches have been proposed for supervised similarity learning using either pairs [25], or triplets [21] of images. However, supervised formulations for learning similarities require that the supervisory information scales quadratically for pairs of images, or cubically for triplets. This results in very large training times.

Literature on exemplar based learning in CNNs is very scarce. In [4] the authors of Exemplar-CNN tackle the problem of unsupervised feature learning. A patch-based categorization problem is designed by randomly extracting patch for each image in the training set and defining it as surrogate class. Hence, since this approach does not take into account (dis-)similarities between exemplars, it fails to model their transitivity relationships, resulting in poor performances (see Sect. 3.1). le

Furthermore, recent works by Wang et al. [22] and Doersh et al. [3] showed that temporal information in videos and spatial context information in images can be utilized as a convenient supervisory signal for learning feature representation with CNNs. However, the computational cost of the training algorithm is enormous since the approach in [3] needs to tackle all possible pair-wise image relationships requiring training set that scales quadratically with the number of samples. On the contrary, our approach leverages the relationship information between compact cliques, defining a multi-class classification problem. As each training batch contains mutually distinct cliques the computational cost of the training algorithm is greatly decreased.

## 2 Approach

We will now discuss how we can employ a CNN for learning similarities between all pairs of a large number of exemplars. Exemplar learning in CNNs has been a relatively unexplored approach for multiple reasons. First and foremost, deep learning requires large amounts of training data, thus

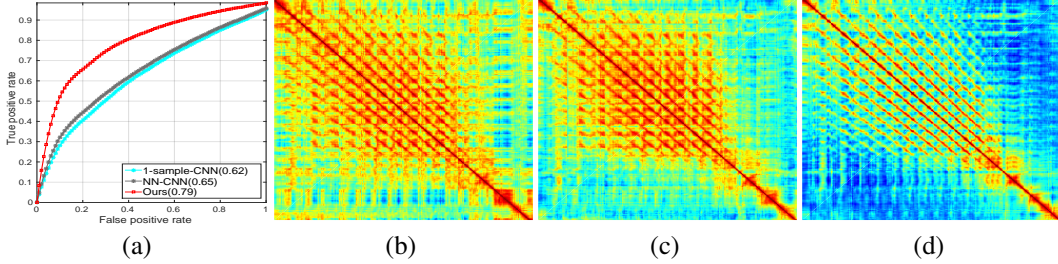

Figure 1: (a) Average AUC for posture retrieval in the Olympic Sports dataset. Similarities learnt by (b) 1-sample CNN, (c) using NN-CNN, and (d) for the proposed approach. The plots show a magnified crop of the full similarity matrix. Note the more detailed fine structure in (d).

conflicting with having only a single positive exemplar in a setup that we now abbreviate as 1-sample CNN. Such a 1-sample CNN faces several issues. *(i)* The within-class variance of an individual exemplar cannot be modeled. *(ii)* The ratio of one exemplar and many negatives is highly imbalanced, so that the softmax loss over SGD batches overfits against the negatives. *(iii)* An SGD batch for training a CNN on multiple exemplars can contain arbitrarily similar samples with different label (the different exemplars may be similar or dissimilar), resulting in label inconsistencies. The proposed method overcomes these issues as follows. In Sect. 2.2 we discuss why simply merging an exemplar with its nearest neighbors and data augmentation (similar in spirit to the Clustered Exemplar-SVM [20]) is not sufficient to address *(i)*. Sect. 3.1 compares this NN-CNN approach against other methods. Sect. 2.3 deals with *(ii)* and *(iii)* by generating batches of cliques that maximize the intra-clique similarity while minimizing inter-clique similarity.

To show the effectiveness of the proposed method we give empirical proof by training CNNs in both 1-sample CNN and NN-CNN manners. Fig. 1(a) shows the average ROC curve for posture retrieval in the Olympic Sports dataset [16] (refer to Sec. 3.1 for further details) for 1-sample CNN, NN-CNN and the proposed method, which clearly outperforms both exemplar based strategies. In addition, Fig. 1(b-d) show an excerpt of the similarity matrix learned for each method. It becomes evident that the proposed approach captures more detailed similarity structures, e.g., the diagonal structures correspond to repetitions of the same gait cycle within a long jump.

## 2.1 Initialization

Since deep learning benefits from large amounts of data and requires more than a single exemplar to avoid biased gradients, we now reframe exemplar-based learning of similarities so that it can be handled by a CNN. Given a single exemplar $\mathbf{d}_i$ we thus strive for related samples to enable a CNN training that then further improves the similarities between samples. To obtain this initial set of few, mutually similar samples for an exemplar, we now briefly discuss the reliability of standard feature distances such as whitening HOG features using LDA [10]. HOG-LDA is a computationally effective foundation for estimating similarities $s_{ij}$ between large numbers of samples, $s_{ij} = s(\mathbf{d}_i, \mathbf{d}_j) = \phi(\mathbf{d}_i)^\top \phi(\mathbf{d}_j)$. Here $\phi(\mathbf{d}_i)$ is the initial HOG-LDA representation of the exemplar and $\mathbf{S}$ is the resulting kernel.

Most of these initial similarities are unreliable (cf. Fig. 4(b)) and, thus, the majority of samples cannot be properly ranked w.r.t. their similarity to an exemplar $\mathbf{d}_i$. However, highly similar samples and those that are far away can be reliably identified as they stand out from the similarity distribution. Subsequently we utilize these few reliable relationships to build groups of compact cliques.

## 2.2 Compact Cliques

Simply assigning the same label to all the nearest and another label to all the furthest neighbors of an exemplar is inappropriate. The samples in these groups may be close to $\mathbf{d}_i$ (or distant for the negative group) but not to another due to lacking transitivity. Moreover, mere augmentation of the exemplar with synthetic data does not add transitivity relations to other samples. Therefore, to learn within-class similarities we need to restrict the model to compact cliques of samples so that all samples in a clique are also mutually close to another and deserve the same label.

| Query | Ours | Alexnet [13] | HOG-LDA [10] |
|---|---|---|---|

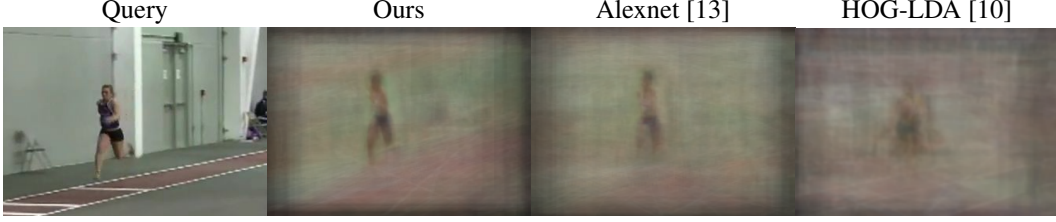

Figure 2: Averaging of the 50 nearest neighbours for a given query frame using similarities obtained by our approach, Alexnet[13] and HOG-LDA [10].

To build candidate cliques we apply complete-linkage clustering starting at each $d_i$ to merge the sample with its local neighborhood, so that all merged samples are mutually similar. Thus, cliques are compact, differ in size, and may be mutually overlapping. To reduce redundancy, highly overlapping cliques are subsequently merged by clustering cliques using farthest-neighbor clustering. This agglomerative grouping is terminated if intra-clique similarity of a cluster is less than half that of its constituents. Let $K$ be the resulting number of clustered cliques and $N$ the number of samples $d_i$. Then $\mathbf{C} \in \{0,1\}^{K \times N}$ is the resulting assignment matrix of samples to cliques.

## 2.3 Selecting Batches of Mutually Consistent Cliques

We now have a set of compact cliques that comprise all training data. Thus, one may consider to train a CNN to assign all samples of a clique with the same label. However, since only the highest/lowest similarities are reliable, samples in different cliques are not necessarily dissimilar. Forcing them into different classes can consequently entail incorrect similarities. Therefore, we now seek batches of mutually distant cliques, so that all samples in a batch can be labeled consistently because they are either similar (same compact clique) or dissimilar (different, distant clique). Samples with unreliable similarity then end up in different batches and we train a CNN successively on these batches.

We now formulate an optimization problem that produces a set of consistent batches of cliques. Let $\mathbf{X} \in \{0,1\}^{B \times K}$ be an indicator matrix that assigns $K$ cliques to $B$ batches (the rows $\mathbf{x}_b$ of $\mathbf{X}$ are the cliques in batch $b$) and $\mathbf{S'} \in \mathbb{R}^{K \times K}$ be the similarity between cliques. We enforce cliques in the same batch to be dissimilar by minimizing $\operatorname{tr}(\mathbf{X}\mathbf{S'}\mathbf{X}^\top)$, which is regularized for the diagonal elements of the matrix $\mathbf{S'}$ selected for each batch (see Eq. (1)). Moreover, each batch should maximize sample coverage, i.e., the number of distinct samples in all cliques of a batch $\|\mathbf{x}_b\mathbf{C}\|_p^p$ should be maximal. Finally, the number of distinct points covered by all batches, $\|\mathbb{1}\mathbf{X}\mathbf{C}\|_p^p$, should be maximal, so that the different (potentially overlapping) batches together comprise as much samples as possible. We select $p = 1/16$ so that our penalty function roughly approximate the non-linear step function. The objective of the optimization problem then becomes

$$\min_{\mathbf{X} \in \{0,1\}^{B \times K}} \operatorname{tr}(\mathbf{X}\mathbf{S'}\mathbf{X}^\top) - \operatorname{tr}(\mathbf{X}\operatorname{diag}(\mathbf{S'})\mathbf{X}^\top) - \lambda_1 \sum_{b=1}^{B} \|\mathbf{x}_b\mathbf{C}\|_p^p - \lambda_2 \|\mathbb{1}\mathbf{X}\mathbf{C}\|_p^p \tag{1}$$

$$\text{s.t.} \qquad \mathbf{X}\mathbb{1}_K^\top = r\mathbb{1}_B^\top \tag{2}$$

where $r$ is the desired number of cliques in one batch for CNN training. The number of batches, $B$, can be set arbitrarily high to allow for as many rounds of SGD training as desired. If it is too low, this can be easily spotted as only limited coverage of training data can be achieved in the last term of Eq. (1). Since $\mathbf{X}$ is discrete, the optimization problem (1) is not easier than the Quadratic Assignment Proble which is known to be $NP$-hardm [1]. To overcome this issue we relax the binary constraints and force instead the continuous solution to the boundaries of the feasible range by maximizing the additional term $\lambda_3 \|\mathbf{X} - 0.5\|_F^2$ using the Frobenius norm.

We condition $\mathbf{S'}$ to be positive semi-definite by thresholding its eigenvectors and projecting onto the resulting base. Since also $p < 1$ the previous objective function is a difference of convex functions

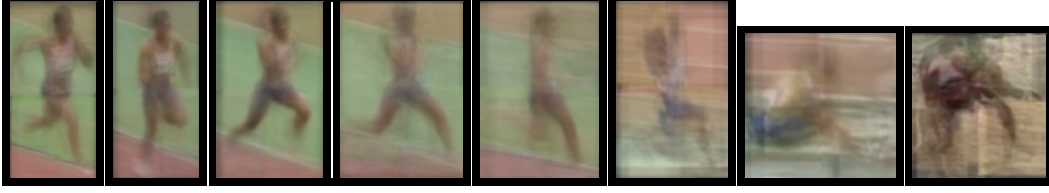

Figure 3: Visual example of a resulting batch of cliques for long jump category of Olympic Sports dataset. Each clique contains at least 20 samples and is represented as their average.

$u(\mathbf{X}) - v(\mathbf{X})$, where

$$u(\mathbf{X}) = \text{tr}\left(\mathbf{X}\mathbf{S}'\mathbf{X}^\top\right) - \lambda_1 \sum_{b=1}^{B} \|\mathbf{x}_b \mathbf{C}\|_p^p - \lambda_2 \|\mathbb{1}\mathbf{X}\mathbf{C}\|_p^p \tag{3}$$

$$v(\mathbf{X}) = \text{tr}(\mathbf{X}\,\text{diag}\,(\mathbf{S}')\mathbf{X}^\top) + \lambda_3 \|\mathbf{X} - 0.5\|_F^2 \tag{4}$$

It can be solved using the CCCP Algorithm [24]. In each iteration of CCCP the following convex optimization problem is solved,

$$\underset{\mathbf{X} \in [0,1]^{B \times K}}{\text{argmin}} \ u(\mathbf{X}) - \text{vec}\,(\mathbf{X})^\top \,\text{vec}\,(\nabla v(\mathbf{X}^t)), \tag{5}$$

$$\text{s.t.} \qquad \mathbf{X}\mathbb{1}_K^\top = r\mathbb{1}_B^\top \tag{6}$$

where $\nabla v(\mathbf{X}^t) = 2\mathbf{X} \odot (\mathbb{1}\,\text{diag}\,(\mathbf{S}')) + 2\mathbf{X} - \mathbb{1}$ and $\odot$ denotes the Hadamard product. We solve this constrained optimization problem by means of the interior-point method. Fig. 3 shows a visual example of a selected batch of cliques.

## 2.4 CNN Training

We successively train a CNN on the different batches $\mathbf{x}_b$ obtained using Eq. (1). In each batch, classifying samples according to the clique they are in then serves as a pretext task for learning sample similarities. One of the key properties of CNNs is the training using SGD and backpropagation [14]. The backpropagated gradient is estimated only over a subset (batch) of training samples, so it depends only on the subset of cliques in $\mathbf{x}_b$. Following this observation, the clique categorization problem is effectively decoupled into a set of smaller sub-tasks—the individual batches of cliques. During training, we randomly pick a batch $b$ in each iteration and compute the stochastic gradient, using the softmax loss $L(\mathbf{W})$,

$$L(\mathbf{W}) \approx \frac{1}{M} \sum_{j \in \mathbf{x}^b} f_{\mathbf{W}}(\mathbf{d}_j) + \lambda r(\mathbf{W}) \tag{7}$$

$$\mathbf{V}_{t+1} = \mu \mathbf{V}_t - \alpha \nabla L(\mathbf{W}_t), \qquad \mathbf{W}_{t+1} = \mathbf{W}_t + \mathbf{V}_{t+1} , \tag{8}$$

where $M$ is the SGD batch size, $\mathbf{W}_t$ denotes the CNN weights at iteration $t$, and $\mathbf{V}_t$ denotes the weight update of the previous iteration. Parameters $\alpha$ and $\mu$ denote the learning rate and momentum, respectively. We then compute similarities between exemplars by simply measuring correlation on the learned feature representation extracted from the CNN (see Sect. 3.1 for details).

## 2.5 Similarity Imputation

By alternating between the different batches, which contain cliques with mutually inconsistent similarities, the CNN learns a single representation for samples from all batches. In effect, this consolidates similarities between cliques in different batches. It generalizes from a subset of initial cliques to new, previously unreliable relations between samples in different batches by utilizing transitivity relationships implied by the cliques.

After a training round over all batches we impute the similarities using the representation learned by the CNN. The resulting similarities are more reliable and enable the grouping algorithm from Sect. 2.2 to find larger cliques of mutually related samples. As there are fewer unreliable similarities,

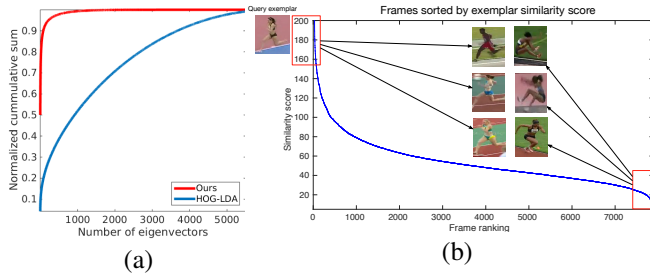

Figure 4: (a) Cumulative distribution of the spectrum of the similarity matrices obtained by our method and the HOG-LDA initialization. (b) Sorted similarities with respect to one exemplar, where only similarities at the ends of the distribution can be trusted.

more samples can be comprised in a batch and overall less batches already cover the same fraction of data as before. Consequently, we alternately train the CNN and recompute cliques and batches using the similarities inferred in the previous iteration of CNN training. This alternating imputation of similarities and update of the classifier follows the idea of multiple-instance learning and has shown to converge quickly in less than four iterations.

To evaluate the improvement of the similarities Fig. 4 analyzes the eigenvalue spectrum of $\mathbf{S}$ on the Olympic Sports dataset, see Sect. 3.1. The plot shows the normalized cumulative sum of the eigenvalues as the function of the number of eigenvectors. Compared to the initialization, transitivity relations are learned and the approach can generalize from an exemplar to more related samples. Therefore, the similarity matrix becomes more structured (cf. Fig. 1) and random noisy relations disappear. As a consequence it can be represented using very few basis vectors. In a further experiment we evaluate the number of reliable similarities and dissimilarities within and between cliques per batch. Recall that samples can only be part of the same batch, if their similarity is reliable. So the goal of similarity learning is to remove transitivity conflicts and reconcile relations between samples to yield larger batches. We now observe that after the iterative update of similarities, the average number of similarities and dissimilarities in a batch has increased by a factor of 2.34 compared to the batches at initialization.

# 3 Experimental Evaluation

We provide a quantitative and qualitative analysis of our exemplar-based approach for unsupervised similarity learning. For evaluation, three different settings are considered: posture analysis on Olympic Sports [16], pose estimation on Leeds Sports [12], and object classification on PASCAL VOC 2007.

## 3.1 Olympic Sports Dataset: Posture Analysis

The Olympic Sports dataset [16] is a video compilation of different sports competitions. To evaluate fine-scale pose similarity, for each sports category we had independent annotators manually label 20 positive (similar) and negative (dissimilar) samples for 1033 exemplars. Note that these annotations are solely used for testing, since we follow an unsupervised approach.

We compare the proposed method with the Exemplar-CNN [4], the two-stream approach of Doersch et. al [3], 1-sample CNN and NN-CNN models (in a very similar spirit to [20]), Alexnet [13], Exemplar-SVMs [15], and HOG-LDA [10]. Due to its performance in object and person detection, we use the approach of [7] to compute person bounding boxes. *(i)* The evaluation should investigate the benefit of the unsupervised gathering of batches of cliques for deep learning of exemplars using standard CNN architectures. Therefore we incarnate our approach by adopting the widely used model of Krizhevsky et al. [13]. Batches for training the network are obtained by solving the optimization problem in Eq. (1) with $B = 100$, $K = 100$, and $r = 20$ and fine-tuning the model for $10^5$ iterations. Thereafter we compute similarities using features extracted from layer fc7 in the *caffe* implementation of [13]. *(ii)* Exemplar-CNN is trained using the best performing parameters reported in [4] and the 64c5-128c5-256c5-512f architecture. Then we use the output of fc4 and compute 4-quadrant max pooling. *(iii)* Exemplar-SVM was trained on the exemplar frames using the HOG descriptor. The samples for hard negative mining come from all categories except the one that an exemplar is from. We performed cross-validation to find an optimal number of negative mining rounds (less than three). The class weights of the linear SVM were set as $C_1 = 0.5$ and $C_2 = 0.01$. *(iv)* LDA whitened HOG

| HOG-LDA [10] | Ex-SVM [15] | Ex-CNN [4] | Alexnet [13] | 1-s CNN | NN-CNN | Doersch et. al [3] | Ours |
|---|---|---|---|---|---|---|---|
| 0.58 | 0.67 | 0.56 | 0.65 | 0.62 | 0.65 | 0.58 | **0.79** |

Table 1: Avg. AUC for each method on Olympic Sports dataset.

was computed as specified in [10]. *(v)* The 1-sample CNN was trained by defining a separate class for each exemplar sample plus a negative category containing all other samples. *(vi)* In a similar fashion, the NN-CNN was trained using the exemplar plus 10 nearest neighbours obtained using the whitened HOG similarities. As implementation for both CNNs we again used the model of [13] fine-tuned for $10^5$ iterations. Each image in the training set is augmented with 10 transformed versions by performing random translation, scaling, rotation and color transformation, to improve invariance with respect to these.

Tab. 1 reports the average AuC for each method over all categories of the Olympic Sports dataset. Our approach obtains a performance improvement of at least $10\%$ w.r.t. the other methods. In particular, the experiments show that the 1-sample CNN fails to model the positive distribution, due to the high imbalance between positives and negatives and the resulting biased gradient. In comparison, additional nearest neighbours to the exemplar (NN-CNN) yield a better model of within-class variability of the exemplar leading to a $3\%$ performance increase over the 1-sample CNN. However NN-CNN also sees a large set of negatives, which are partially similar and dissimilar. Due to this unstructuredness of the negative set, the approach fails to thoroughly capture the fine-grained similarity structure over the negative samples. To circumvent this issue we compute sets of mutually distant compact cliques resulting in a relative performance increase of $12\%$ over NN-CNN.

Furthermore, Fig. 1 presents the similarity structures, which the different approaches extract when analyzing human postures. Fig. 2 further highlights the similarities and the relations between neighbors. For each method the top 50 nearest neighbours for a randomly chosen exemplar frame in the Olympic Sports dataset are blended. We can see how the neighbors obtained by our approach depict a sharper average posture, since they result from compact cliques of mutually similar samples. Therefore they retain more details and are more similar to the original than in case of the other methods.

## 3.2   Leeds Sports Dataset: Pose Estimation

The Leeds Sports Dataset [12] is the most widely used benchmark for pose estimation. For training we employ 1000 images from the dataset combined with 4000 images from the extended version of this dataset, where each image is annotated with 14 joint locations. We use the visual similarities learned by our approach to find frames similar in posture to a query frame. Since our training is unsupervised, joint labels are not available. At test time we therefore estimate the pose of a query person by identifying the nearest neighbor from the training set. To compare against the supervised methods, the pose of the nearest neighbor is then compared against ground-truth.

Now we evaluate our visual similarity learning and the resulting identification of nearest postures. For comparison, similar postures are also retrieved using HOG-LDA [10] and Alexnet [13]. In addition, we also report an upper bound on the performance that can be achieved by the nearest neighbor using ground-truth similarities. Therefore, the nearest training pose for a query is identified by minimizing the average distance between their ground-truth pose annotation. This is the best one can do by finding the most similar frame, when not provided with a supervised parametric model (the performance gap to $100\%$ shows the difference between training and test poses). For completeness, we compare with a fully supervised state-of-the-art approach for pose estimation [18]. We use the same experimental settings described in Sect. 3.1.

Tab. 2 reports the Percentage of Correct Parts (PCP) for the different methods. The prediction for a part is considered correct when its endpoints are within $50\%$ part length of the corresponding ground truth endpoints. Our approach significantly improves the visual similarities learned using Alexnet and HOG-LDA. It is note-worthy that even though our approach for estimating the pose is *fully unsupervised* it attains a competitive performance when compared to the upper-bound of supervised ground truth similarities.

In addition, Fig. 5 presents success (a) and failure (c) cases of our method. In Fig.5(a) we can see that the pose is correctly transferred from the nearest neighbor (b) from the training set, resulting in a PCP score of 0.6 for that particular image. Moreover, Fig.5(c), (d) show that the representation learnt

| Method | Torso | Upper legs | Lower legs | Upper arms | Lower arms | Head | Total |
|---|---|---|---|---|---|---|---|
| Ours | 80.1 | 50.1 | 45.7 | 27.2 | 12.6 | 45.5 | 43.5 |
| HOG-LDA[10] | 73.7 | 41.8 | 39.2 | 23.2 | 10.3 | 42.2 | 38.4 |
| Alexnet[13] | 76.9 | 47.8 | 41.8 | 26.7 | 11.2 | 42.4 | 41.1 |
| Ground Truth | 93.7 | 78.8 | 74.9 | 58.7 | 36.4 | 72.4 | 69.2 |
| Pose Machines [18] | 93.1 | 83.6 | 76.8 | 68.1 | 42.2 | 85.4 | 72.0 |

Table 2: PCP measure for each method on Leeds Sports dataset.

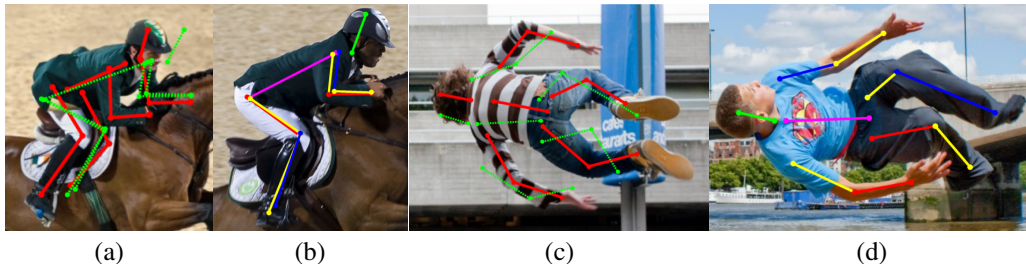

(a)  (b)  (c)  (d)

Figure 5: Pose prediction results. (a) and (c) are test images with the superimposed ground truth skeleton depicted in red and the predicted skeleton in green. (b) and (d) are corresponding nearest neighbours, which were used to transfer pose.

by our method is invariant to front-back flips (matching a person facing away from the camera to one facing the camera). Since our approach learns pose similarity in an unsupervised manner, it becomes invariant to changes in appearance as long as the shape is similar, thus explaining this confusion. Adding additional training data or directly incorporating face detection-based features could resolve this.

### 3.3 PASCAL VOC 2007: Object Classification

The previous sections have analyzed the learning of pose similarities. Now we evaluate the learning of similarities over object categories. Therefore, we classify object bounding boxes of the PASCAL VOC 2007 dataset. To initialize our model we now use the visual similarities of Wang et al. [22] without applying any fine tuning on PASCAL and also compare against this approach. Thus, neither ImageNet nor Pascal VOC labels are utilized. For comparison we evaluate against HOG-LDA [10], [22], and R-CNN [9]. For our method and HOG-LDA we use the same experimental settings as described in Sect. 3.1, initializing our method and network with the similarities obtained by [22]. For all methods, the $k$ nearest neighbors are computed using similarities (Pearson correlation) based on fc6. In Tab. 3 we show the classification accuracies for all approaches for $k = 5$. Our approach improves upon the initial similarities of the unsupervised approach of [22] to yield a performance gain of 3% without requiring any supervision information or fine-tuning on PASCAL.

| HOG-LDA | Wang et. al [22] | Wang et. al [22] + Ours | RCNN |
|---|---|---|---|
| 0.1180 | 0.4501 | 0.4812 | 0.6825 |

Table 3: Classification results for PASCAL VOC 2007

## 4   Conclusion

We have proposed an approach for unsupervised learning of similarities between large numbers of exemplars using CNNs. CNN training is made applicable in this context by addressing crucial problems resulting from the single positive exemplar setup, the imbalance between exemplar and negatives, and inconsistent labels within SGD batches. Optimization of a single cost function yields SGD batches of compact, mutually dissimilar cliques of samples. Learning exemplar similarities is then posed as a categorization task on individual batches. In the experimental evaluation the approach has shown competitive performance compared to the state-of-the-art, providing significantly finer similarity structure that is particularly crucial for detailed posture analysis.

This research has been funded in part by the Ministry for Science, Baden-Württemberg and the Heidelberg Academy of Sciences, Heidelberg, Germany. We are grateful to the NVIDIA corporation for donating a Titan X GPU.

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
