[Supplementary Material]

# Supplementary Material for "CliqueCNN: Deep Unsupervised Exemplar Learning"

**Miguel A. Bautista**[*]**, Artsiom Sanakoyeu**[*]**, Ekaterina Sutter, Björn Ommer**
Heidelberg Collaboratory for Image Processing
IWR, Heidelberg University, Germany
`firstname.lastname@iwr.uni-heidelberg.de`

## 1 Appendix

### 1.1 ROCs for Olympic Sports

In this annex we show ROC curves for each individual category of Olympic Sports in Fig. 1, demonstrating that the improvements obtained by our approach are consistent over all classes. In particular, the experiments show that the 1-sample CNN fails to model the positive distribution, due to the high imbalance between positives and negatives and the resulting biased gradient. In comparison, additional nearest neighbours to the exemplar (NN-CNN) yield a better model of within-class variability of the exemplar leading to a $3\%$ performance increase over the 1-sample CNN. However NN-CNN also sees a large set of negatives, which are partially similar and dissimilar. Due to this unstructuredness of the negative set, the approach fails to thoroughly capture the fine-grained similarity structure over the negative samples. To circumvent this issue, CliqueCNN computes sets of mutually distant compact cliques, resulting in a relative performance increase of $12\%$ over NN-CNN.

### 1.2 Qualitative Results

In this section we present qualitative retrieval results for Olympic Sports in Fig. 2, Leeds Sports in Fig. 3 and VOC2007 datasets in Fig. 4.

---

[*]Both authors contributed equally

Figure 1: ROC curves for Olympic Sports dataset where AUC for each method is reported between parenthesis on the legend of each figure. (a) Basketball layup. (b) Bowling. (c) Clean and jerk. (d) Discus throw. (e) Diving platform 10m. (f) Diving springboard 3m. (g) Hammer throw. (h) High jump. (i) Javelin throw. (j) Long jump. (k) Pole vault. (l) Shot put. (m) Snatch. (n) Tennis serve. (o) Triple jump. (p) Vault.

**Query** | **NNs**

Figure 2: Nearest neighbours retrieved by CliqueCNN for representative query images of the Olympic Sports dataset.

| Query | NNs |
|-------|-----|

Figure 3: Nearest neighbours retrieved by CliqueCNN for representative query images of the Leeds Sports dataset.

Figure 4: Nearest neighbours retrieved by CliqueCNN for representative query images of the VOC2007 dataset.