[Reviews · NeurIPS 2016]

Reviewer 1

Summary

The manuscript studies the task of unsupervised construction of features for similarity determination from exemplars. It proposes a new training paradigm and corresponding loss function [a differentiable version of a discrete clustering loss] wherein a CNN architecture is combined with clustering algorithms to allow it to circumvent the problem of highly imbalanced training data when using exemplars with large neural networks. They demonstrate the performance of the approach on three standard datasets: Olympic images, Pose estimation, and the Pascal object recognition dataset.

Qualitative Assessment

The paper proposes an interesting new way of using CNNs to learn unsupervised features of similarity from exemplars. It develops a novel differentiable loss and uses clustering to break data into mutually distinct cliques that allow balancing of the data (so that the usual problem of a small number of exemplars versus many negative examples might be mitigated). In general the paper is well written and motivated and builds nicely on the trend of combining discrete optimization algorithms with deep networks for compelling effects. The results across several real-world datasets look promising, though a bit more could be done to strengthen these. One worry I did have: Why switch to a different initialization of features for the PASCAL dataset (i.e. from HOG-LDA to Wang et al.)? Do the HOG-LDA features not work well for initialization in this case? Can the proposed approach recover from such a bad initialization? Minor Issues: The phrase "gets corrupted" on line 41 seems an odd wording. Line 130 could perhaps use a citation for complete-linkage clustering. For PASCAL the results are reported only for k=5. It would be nice to know the figures for k=1,...,10. Is the 3% boost contingent on having k=5? Grammar/Spelling issues on lines: 2 -- there's something a bit funny with the tense here. 76 -- "extracting patch for..." 126 -- "but not to another due to..." 225 -- "by solve"

Confidence in this Review

2-Confident (read it all; understood it all reasonably well)


Reviewer 2

Summary

This paper presents an approach for unsupervised learning of similarities between exemplars. The paper addresses the problem of selecting relevant positive examples and negative examples for CNN-based similarity training. The underlying approach has two steps. First, an optimization problem is solved to extract samples with mutually consistent relations. Then, a ConvNet learns a single representation for all the samples. This procedure is repeated -- the authors alternatively train the CNN and recompute cliques/batches using similarities inferred from the last iteration. Results are shown on sports-related action datasets as well as a traditional PASCAL image classification task.

Qualitative Assessment

Overall, the paper does a good job introducing an iterative algorithm for exemplar-based similarity learning which uses CNNs. The proposed algorithm overcomes some of the key problems when working with exemplar "single-positive" methods. Pros -Algorithm does well compared to baselines Cons -The computational complexity of the "mutually consistent relations" algorithm is not discussed. -It is not clear whether the algorithm can be improved by selecting better mutually consistent cliques, or doing better CNN learning. -Figures 1 and 4 are not clear.

Confidence in this Review

2-Confident (read it all; understood it all reasonably well)


Reviewer 3

Summary

This paper proposes an unsupervised learning method for deep networks based on classifying samples into a pre-defined set of cliques. An optimization procedure is proposed to partition the samples into batches that respect the known similarity and dissimilarity relationships between the clique samples, and leave out pairs with unknown relationship. The method is iterated to refine the learned similarity network, starting with HOG-LDA similarity in the first iteration. Results are demonstrated by nearest-neighbor classification using the learned similarity in a number of datasets.

Qualitative Assessment

The paper reads well and is technically sound, especially the well-defined optimization problem in Sec. 2.3. The idea of consolidating transitivity relationships into batches is an interesting angle in similarity learning. Excluding Section 2.3, however, the method as a whole feels a bit ad-hoc. There doesn't seem to be any reason why the assignment to batches could not be performed on samples directly, rather than cliques (other than that it might speed things up). It would make for a more well-justified method to introduced it in terms of the raw individual samples. Using cliques instead of samples could then be presented as an engineering trick to reduce the problem size. The starting point of the paper seems to be that initial similarities are unreliable, and so there should be no label (training signal) assigned to most of them. It would be more satisfying to relate the optimization objective to this goal explicitly, framing it as an approximation of the full problem if possible. The paper's presentation is good overall. The description in Section 2.2 is too terse, however, making reproducibility suffer; it is also missing supporting references (e.g. "complete-linkage clustering"). The experiments are adequate, especially the fact that the authors used the learned similarities for nearest-neighbor classification/prediction in challenging tasks. The comparisons with ground-truth nearest-neighbors and supervised methods are especially illuminating, placing the unsupervised learning results into perspective. However, there could be a better experimental breakdown of the relative contributions of the aspects of the pipeline, since it is hard to untangle the different components. Simply training a classifier on cliques (suggested in line 141, "Forcing them into different classes ...") would be a good measure of the influence of the cliques clustering on the final result. Another important baseline would be training a simple two-stream network end-to-end on the same data. The authors claim that these two approaches would incorporate incorrect relationships, so it is important to show how the proposed method helps in that regard. It is also debatable whether visualizing merged RGB images really shows that the similarities are meaningful; it is possible there is high pixel-level similarity, but not necessarily semantic similarity. For example, it would be possible for clustering based on RGB values alone to obtain similar visualizations. Another option would be to compute objective measures, like cluster purity, to complement and validate these results.

Confidence in this Review

2-Confident (read it all; understood it all reasonably well)


Reviewer 4

Summary

This paper discusses an approach for examplar learning, based on deep learning techniques. In particular, the problem where we have only a single positive example for a class, and many negative examples. A method for selecting batches for CNN training is proposed. The experiments show competitive results.

Qualitative Assessment

L71-L73: "However, supervised formulations for learning similarities require that the supervisory information scales quadratically for pairs of images, or cubically for triplets. This results in very large training times." => Since CNNs are trained with minibatches, the training time doesn't necessarily scale linearly with number of tuples/triples. The training time depends on the (conditional) information content in the data, which typically grows sublinearly with the amount of data points. L93: "(ii) The ratio of one exemplar and many negatives is highly imbalanced, so that the softmax loss over SGD batches overfits against the negatives." L110-11: "Since deep learning benefits from large amounts of data and requires more than a single exemplar to avoid biased gradients" => Imbalance/bias can be easily fixed through importance-weighted gradients and/or a reweighted objective. The paper is clearly written, but in my opinion does not give a sufficiently clear introduction into examplar-based learning. At the end of page 3 it's still not entirely clear to me what the exact problem is that this paper is attacking. The experiments show promising results on posture analysis and pose estimation. Due to my unfamiliarity with the problem and related literature, I can not reliably assess the quality of contributions of this paper.

Confidence in this Review

1-Less confident (might not have understood significant parts)


Reviewer 5

Summary

Starting with some simple similarity measure, the paper proposes to improve the quality of similarities though an iterative process. In each iteration they first create the cliques (small similar groups) of samples and then identify good training batches by solving an optimization problem. Then using these training batches a CNN is trained. The resulting similarities are mainly tested on human pose similarity where is shows competitive results. They also included a similarity test on PASCAL VOC dataset.

Qualitative Assessment

Even though the technical parts (the formulations) are reasonable, the introduction of the problem and the steps are not too clear. It would be great if the authors can provide a clear big picture for the problem and then take each step individually and explain in detail. The method appears a bit ad-hoc in the sense that it has many components but the effect of each component is not evaluated individually. The selection of each component could be better motivated through comparing it with some (simpler) alternatives. For instance, it would be helpful to see random batch assignments (or some simpler method) versus the proposed batch assignment on at least one of the tasks. Experimental validation shows some promising results on unsupervised pose similarity learning.

Confidence in this Review

2-Confident (read it all; understood it all reasonably well)


Reviewer 6

Summary

This paper presents an unsupervised algorithm for training deep neutral networks. It is based on exemplar-based methods for similarity learning, but tries to tackle a few intrinsic limitations of this type of approach (line #91-101). The proposed algorithm alternates between (1) partitioning dataset into batches of compact cliques; and (2) optimizing the neutral network with a softmax loss on pseudo clique labels. Experiments on posture analysis and pose estimation problems demonstrate great performance, which is even competitive to fully supervised approach in some cases.

Qualitative Assessment

The proposed method is technically sound. With "offline" batch / clique data partitioning, it seems to well solve the problems of traditional exemplar-based approaches. This advantage can be seen from the compelling results in experiments section. The only concern of mine is that whether such an alternating approach would make the learning process less efficient, or less automatic (if extra hyper-parameter tuning is required across iterations / epochs). I'd love to see some discussion on these matters, as well as how the network performance improves over iterations / epochs. The presentation can be improved, e.g. - What are f() and r() in Eq. (7)? - (line #93) The ratio of one exemplar and many negatives is highly imbalanced ... - (line #88) ... learning similarities between all pairs of a large number of exemplars

Confidence in this Review

1-Less confident (might not have understood significant parts)